# Rolling-Circle Replication in Mitochondrial DNA Inheritance: Scientific Evidence and Significance from Yeast to Human Cells

**DOI:** 10.3390/genes11050514

**Published:** 2020-05-06

**Authors:** Feng Ling, Minoru Yoshida

**Affiliations:** 1Chemical Genetics Research Group, RIKEN Center for Sustainable Resource Science, Hirosawa 2-1, Wako, Saitama 351-0198, Japan; yoshidam@riken.jp; 2Graduate School of Science and Engineering, Saitama University, Saitama 338-8570, Japan; 3Department of Biotechnology, Graduate School of Agricultural Life Sciences, the University of Tokyo, Tokyo 113-8657, Japan; 4Collaborative Research Institute for Innovative Microbiology, the University of Tokyo, Tokyo 113-8657, Japan

**Keywords:** homologous recombination, rolling-circle replication, concatemers, oxidative stress, homoplasmy, heteroplasmy

## Abstract

Studies of mitochondrial (mt)DNA replication, which forms the basis of mitochondrial inheritance, have demonstrated that a rolling-circle replication mode exists in yeasts and human cells. In yeast, rolling-circle mtDNA replication mediated by homologous recombination is the predominant pathway for replication of wild-type mtDNA. In human cells, reactive oxygen species (ROS) induce rolling-circle replication to produce concatemers, linear tandem multimers linked by head-to-tail unit-sized mtDNA that promote restoration of homoplasmy from heteroplasmy. The event occurs ahead of mtDNA replication mechanisms observed in mammalian cells, especially under higher ROS load, as newly synthesized mtDNA is concatemeric in hydrogen peroxide-treated human cells. Rolling-circle replication holds promise for treatment of mtDNA heteroplasmy-attributed diseases, which are regarded as incurable. This review highlights the potential therapeutic value of rolling-circle mtDNA replication.

## 1. Introduction

The mitochondria, which provide eukaryotic cells with energy through oxidative phosphorylation, contain multiple copies of mitochondrial DNA (mtDNA). mtDNA encodes components essential for ATP production [1,2]. Mitochondria are the primary intracellular source of reactive oxygen species (ROS), which damage cellular components such as mtDNA molecules [3]. Consequently, mtDNA is more susceptible to mutagenesis than nuclear chromosomal DNA [4,5]. Homoplasmy of mtDNA, in which all mtDNA copies have identical sequences, is the basic state in cells and individuals. A substantial shift toward homoplasmy occurs in fetuses, as well as in embryonic stem cells (ESCs) derived from heteroplasmic embryos [6]. The return of a heteroplasmic mtDNA mutation to homoplasmy occurs in only two or three generations of Holstein cows [7,8], although the underlying mechanism remains highly controversial [9,10,11,12]. It is partially because the mechanism for mtDNA replication in mammals remains unclear. In mammals, a strand-displacement mechanism was proposed based on circular replicative intermediates observed under an electron microscope [13], which was further refined as a model for replication of animal mtDNA [14]. So far, two asynchronous (strand displacement and ribonucleotide incorporation throughout the lagging strand (RITOLS) [15]) and one synchronous (*strand coupled*) replication models have been proposed for human mtDNA replication. In the asynchronous models, replication from the origin in the H-chain starts earlier, so that the replication of the two chains ends at different times. The synchronous model is more traditional and implies two replication forks with leading and lagging strands initiated at the same origin [16]. For details, please refer to the review article [17]. Recently, the rolling-circle mtDNA replication mechanism in hydrogen peroxide-treated human cells, which promotes mt-allele segregation towards mt-homoplasmy, was revealed [18].

mtDNA mutations accumulate during aging, particularly in nerve and muscle cells, resulting in heteroplasmy, a state in which wild-type and mutant mtDNA molecules co-exist within a cell. When the proportion of pathogenic mutant mtDNA exceeds some threshold, heteroplasmy can cause incurable disease due to mitochondrial dysfunction [19]. Induced pluripotent stem (iPS) cells from elderly patients contain elevated amounts of mutant mtDNA, which may jeopardize efficacy and hold back future iPS cell therapies [20,21,22]. In recent years, the expression of mitochondria-targeted transcription activator-like effector nucleases (TALENs) to cleave pathogenic mtDNA mutations raises the possibility that these mitochondrial nucleases can be therapeutic for some mitochondrial diseases, but it is essential to introduce exogenous factors into heteroplasmic cells for the removal of specific mtDNA [23,24,25]. The methods for decreasing the proportion of mutant mtDNA would be crucial for the treatment of heteroplasmy-induced disorders and future research and development of new stem-cell therapies.

Homologous recombination is a type of reshuffling of genetic information in which two similar or identical DNA sequences are exchanged. Its primary role is to repair double-stranded DNA breaks (DSBs). It is widely believed that mitochondria are descended from endosymbiotic bacteria [26]. In bacteria such as *Escherichia coli*, homologous recombination–dependent DNA replication proceeds by the *θ*-type mode or rolling-circle mode of DNA synthesis, yielding closed-circular DNA monomers or linear-stranded DNA multimers, respectively [27]. It remains unknown which DNA replication mode is preferred in mitochondria, and thus forms the basis of mtDNA inheritance [14,28,29]. 

This article introduces the idea that in order to adapt to the environment inside the mitochondrion, mtDNA replicates via a rolling-circle rather than a *θ*-type mode. Homologous DNA pairing mediates both ROS-stimulated rolling-circle mtDNA replication, which promotes mitochondrial allele segregation toward homoplasmy, and homologous DNA recombination, which is crucial for the repair of harmful mtDNA double-stranded breaks (DSBs), and thus for maintenance of cellular respiration.

## 2. The Origin of *θ*-Type mtDNA Replication

The physical study of isolated mtDNA molecules began in the 1960s using mtDNA purified from yeast and mammalian cells. This initial work revealed mtDNA as a minor band with a density lower than that of the nuclear DNA in cesium chloride density gradient [30]. Electron micrographs of purified mtDNA from mouse fibroblasts show this circularity as a ring structure, which is distinct from the low percentage of linear fragments in purified DNA; the virtual absence of free ends in lysed preparations indicates that a ring structure is the primary, but not necessarily the only, form of mtDNA in vivo [30]. Based on these findings, animal mtDNA was proposed to replicate via a *θ*-type mode [14]: the replication intermediates observed in the electron microscopy images resemble the Greek character “*θ*”, and the circularity of mtDNA supported the idea of *θ*-type replication. 

Recently, however, a rolling-circle replication mechanism producing multimeric lariats of mitochondrial DNA was observed in *Caenorhabditis elegans*, revealing that some animal mtDNAs also use this mode [29]. 

## 3. The Main Problems of *θ*-Type Replication Mode for the Explanation of the Rapid Segregation of mt-Alleles towards Homoplasmy

Following mating of *Saccharomyces cerevisiae* haploid cells, the resultant diploids contain 50–100 copies of mtDNA [31], which segregate to homoplasmy within 20 generations [32,33]. However, *θ*-type replication could not achieve such rapid segregation; in this mode, one template yields one copy, and mathematic modeling simulation has shown that random segregation could yield homoplasmy only when six or fewer mtDNA copies are present in the diploid (i.e., three in each haploid cell) [34]. Rapid segregation of mt-alleles from heteroplasmy towards homoplasmy occurs within a few generations during early oogenesis in metazoan female germlines [8,12,35]. This has also been explained as a bottleneck phenomenon [10,11], which again cannot be explained by the *θ*-type replication mode, as the number of segregating units of mtDNA in mice is still ~200 [35]. Notably in this regard, a rolling-circle replication mechanism produces the multimeric lariats of mtDNA observed in another animal model, *C. elegans* [29]. The mtDNA of tube-dwelling anemone, possibly the longest mitochondrial genome observed to date, is also replicated via a rolling-circle mode [36]. It seemed likely an alternate replication mode could exist, and further research to understand the full picture of mtDNA replication systems of humans and other mammals is required [17]. 

## 4. Why Linear Double-Stranded mtDNA Is Undetectable

Why have only circular mtDNA molecules been visualized in mammalian cells by electron microscopy? Very recently, Peeva et al., reported that linear mtDNA is rapidly degraded by components of the replication machinery in human embryonic kidney cells; specifically, degradation by exonucleases such as the mitochondrial 5′→3′ exonuclease MGME1 and the mitochondrial 3′→5′ exonuclease of mtDNA polymerase POLG eliminates linear mtDNA molecules [37]. Indeed, MGME1-deficient mice accumulate long linear subgenomic mtDNA species [38]. Therefore, in mammalian cells, exonucleases rapidly degrade linear mtDNA synthesized via Strand-coupled DNA replication (SCD replication); only the remaining circular mtDNA molecules can be observed by microscopy; dissimilarly, it is relatively easy to detect a lot of intermediates involved in mtDNA replication in *S. cerevisiae* (Figure 1). The major species of the budding yeast *S. cerevisiae* mtDNA are linear head-to-tail multimers of genomic unit DNA with variable sizes, termed concatemers [39,40]. Concatemers can be formed through rolling circle DNA replication or homologous DNA recombination. Rolling-circle DNA replication is initiated through recombination-dependent mechanisms in some DNA replication systems, such as bacteriophage lambda at the late stage of infection [41], plasmids [42], and even the chromosome in SOS-induced *E. coli* cells [43,44]. In the later stages of λ infection, the DNA replication switches from a theta (θ) mode to a rolling circle (σ) mode, and this switch requires the proteins encoded by the *redα* (λ exonuclease) and *redβ* (β protein) genes required for homologous DNA recombination. In phage T4 of *E. coli*, concatemers are formed through homologous DNA recombination [45]. The rolling circle replication can sustainably produce linear tandem multimers linked by head-to-tail unit-sized mtDNA, (concatemers) using circular mtDNA molecules resistant to the degradation by exonucleases as templates. *S. cerevisiae* petite mutants are respiration-deficient cells, which are unable to grow on media containing only non-fermentable carbon sources (such as glycerol or ethanol) and form small colonies when grown in the presence of fermentable carbon sources (such as glucose), and contain mtDNA with a large deletion or tandem arrays of a mtDNA segment [46]. mtDNA deletion-attributed dysfunctional mitochondria can serve as a signaling platform to promote the loss of redox homeostasis and ROS accumulation [47]. More accumulation of concatemers in yeast petite mutant cells is very likely that the excision-repair enzyme Ntg1 recognizes the bases oxidized by ROS and introduces a DSB at the mtDNA replication origin *ori5* to initiate the rolling-circle mtDNA replication [46,48]. Comparative analysis has revealed that the enzymatic activities involved in mtDNA replication of mammals and yeast are very similar [49], implying that the products or intermediates of rolling-circle replication might be present in mammalian cells, as well as in yeast.

## 5. Evidence for Human mtDNA Recombination

mtDNA recombination occurs in human cells [50,51], although the precise machinery involved remains elusive. For example, mtDNA recombination occurs in humans [51], and its intermediates, such as the four-way (Holliday) junctions observed in the human heart muscle, are sensitive to treatment with *E. coli* RuvC protein, a Holliday junction resolvase [52,53]. 

When pulsed-field gel electrophoresis (PFGE) is used to separate human mtDNA species from nuclear genomic DNA species followed by Southern blot analysis with an mtDNA-specific probe, the majority of mtDNA molecules observed as mtDNA signals remain stuck inside the wells [18]. This is reminiscent of the recombination-mediated replication mechanisms in yeast [40] and plants [54].

## 6. The Rolling-Circle mtDNA Replication Mode is Universal

mtDNA recombination was first observed more than 50 years ago in budding yeast [55], over 25 years ago in plants [56], and over 20 years ago in human cells [50,51]. Homologous recombination is essential for initiation of rolling-circle mtDNA replication, which was first observed in budding yeast [39,57]. Recently, rolling-circle mtDNA replication was proven to be the predominant form in yeast [58], and it also occurs in nematodes [29], plants [59,60], and humans [18]. The research history of mtDNA recombination and rolling-circle mtDNA replication in budding yeast led us to infer that mtDNA recombination events are tightly linked with rolling-circle replication.

## 7. The Mhr1-Driven Mechanism of Rolling Circle mtDNA Replication in Yeast

In budding yeast [61], linear mtDNA molecules, observed as the primary form of mtDNA, declared the end of circle form for yeast mtDNA [28]. These linear mtDNA molecules are mainly linear tandem multimers linked by head-to-tail unit-sized mtDNA, termed mtDNA concatemers, which are produced by rolling-circle replication [33,62]. Although circular mtDNA molecules are a minority of budding yeast mtDNA molecules [40], circular mtDNA can be generated from an event termed intramolecular recombination [63], in which Cce1, a cruciform cutting endonuclease, resolves Holiday junctions as recombination intermediates [64]. mtDNA concatemers in mother cells are likely processed to monomers in buds [33].

*Saccharomyces cerevisiae MHR1*, which encodes the Mhr1 protein, is the wild-type gene complementing a recessive nuclear mutation (*mhr1-1*) that causes a defect in mtDNA recombination [57]. Mhr1, a mitochondrial recombinase [62,65], plays a role in the repair of oxidatively damaged mtDNA [66] and is responsible for initiating rolling-circle mtDNA replication through homologous DNA recombination intermediates termed heteroduplex joints [48,62,67]. The products of such a mtDNA replication mode are linear tandem multimers linked by head-to-tail unit-sized mtDNA, termed mtDNA concatemers [33]. Concatemers enable transmission of multiple identical mt-genome copies as a single unit, and thus promote the segregation of all mitochondrial alleles (i.e., separate sets of normal and mutated variants of each gene) to restore homoplasmy [33]. In the rolling-circle mode, the mitochondrial recombinase Mhr1 mediates homologous DNA pairing to initiate the rolling-circle mtDNA replication (Figure 1). In addition, Mhr1 can bind mtDNA double-strand breaks (DSBs) and mediate homologous DNA recombination, the predominant pathway for repair of mtDNA DSBs [67,68]. Several factors that collaborate with Mhr1 have been identified. For example, the DNA damage–inducible 5′→3′ exonuclease Din7 acts in DNA end resection to produce 3′-single-stranded DNA tails [67]. A mitochondria-localized Rad52-related protein Mgm101 has a short carboxyl-terminal tail for single-stranded DNA binding required for mitochondrial DNA recombination to maintain yeast mtDNA [69,70,71,72]. The oxidized base excision-repair enzyme Ntg1 introduces a DSB in the single-stranded regions at the mtDNA replication origin *ori5*; this DSB initiates the rolling-circle mtDNA replication mediated by Mhr1 [46,48]. Thus, optimal amounts of ROS promote mt-allele segregation mediated concatemers produced by the rolling-circle mtDNA replication, leading us to propose a mechanism in which an optimal level of ROS activates the homologous DNA pairing-initiated recombination-driven rolling-circle replication (RdRR) to increase mtDNA copy number in budding yeast [46,48]. 

ROS can damage DNA, but also serve as central hubs in cellular signaling networks [73]. Mitochondrial ROS stabilizes HIF-1 (hypoxia-inducible factor), a master regulator of hypoxia-induced gene expression [74,75]. Adaptive mitochondrial ROS signal extends the chronological lifespans of both *C. elegans* and yeast [76,77]. Elevated levels of oxidative stress decrease the level of the transcription regulator BACH1, which stimulates lung cancer metastasis [78]. In addition, mitochondrial ROS regulates thermogenic energy expenditure by promoting sulfenylation of uncoupling protein 1 (UCP-1) in brown adipose tissue [79]. 

## 8. Roles of RdRR in Mitochondrial Dynamics and Maintenance of mtDNA Integrity

Mitochondrial nucleoids are regarded as the segregation unit for mtDNA inheritance [80]. Over 50 nucleoid-associated proteins, including aconitase, a component of the TCA cycle, play roles in mtDNA maintenance and gene expression [80,81]. Abf2 (the yeast homolog of human TFAM), a key component of the nucleoid with a histone-like role [82,83], promotes efficient packaging of linear double-stranded DNAs such as concatemers [82] by wrapping and bending mtDNA to protect it from damage and digestion by nucleases [84,85]. Abf2 plays no transcriptional role in yeast [86,87,88]. Mutants lacking *ABF2* (*∆abf2*) lose mtDNA phenotype due to mtDNA deletions [80,82,89,90,91] in yeast and human cells, mtDNA concatemers produced by RdRR are likely packaged by Abf2 or TFAM into a nucleoprotein complex termed the mitochondrial nucleoid (Figure 2). Indeed, activation of the checkpoint via the ATM-Chk2 pathway in response to DNA damage increases mtDNA content without changing the amount of TFAM, but is accompanied by generation of the common 4977-bp deletion [92]. Fusion events, which are accompanied by the degradation of dissociated electron transport chain complex IV and transient reductions in the levels of complex IV subunits, increase ROS levels, leading to elevation of mtDNA copy number in a manner dependent on Mhr1 [93]. Therefore, RdRR ensures the distribution of mitochondrial genomes and is thus critical for the maintenance of mtDNA copy number during mitochondrial dynamics [93]. 

In contrast to Mhr1, overproduction of the Abf2 leads to mtDNA instability [89]. Mhr1 localizes near DSB sites in mtDNA [68], and exogenous introduction of Mhr1 promotes mtDNA recombination, thereby preventing mtDNA deletion-attributed deficiency in respiratory function [91]. This is consistent with the conclusion that Mhr1 plays a pivotal role in mtDNA maintenance [94]. 

As in *S. cerevisiae*, mtDNA replication is initiated from linear double-stranded mtDNA, and circular mtDNA molecules result from intramolecular recombination of linear double-stranded mtDNAs mediated by direct repeats. Circular mtDNA molecules act as templates for rolling-circle mtDNA replication for production of mtDNA concatemers (also see: Figure 1). mtDNA molecules that fail to package in nucleoids are usually sensitive to DNA damages and digestion by nucleases, and consequently generate DSBs. DSBs of mtDNA are repaired by homologous recombination, thereby preventing mtDNA deletions (Figure 2). 

## 9. Significance of the mtDNA Recombination-Driven Rolling-Circle mtDNA Replication

Aging is caused by multiple internal and external factors. mtDNA deletion jeopardizes the ability of mitochondria to provide sufficient ATP, and thus causes processes related to aging [95,96]. In yeast, mtDNA deletion levels negatively correlate with the capacity of mtDNA recombination, in which the rolling-circle type mtDNA replication increases mtDNA copy number while preventing heteroplasmy due to mtDNA deletions [91], allowing us to infer that proficiency of mtDNA recombination is tightly associated with a healthy lifespan. 

In heteroplasmic human cells, the activation of rolling-circle mtDNA replication represents a potential strategy for treating incurable diseases attributed to mitochondrial dysfunction. mtDNA mutations accumulate during aging, particularly in nerve and muscle cells, resulting in heteroplasmy [95,97,98,99,100,101,102]. A common mtDNA point mutation, which is the A-to-G transition at nucleotide position (np) 3243 (m.3243A > G), forms stable heteroplasmy with wild-type mtDNA [103,104] and causes mitochondrial myopathy, encephalopathy, lactic acidosis, and stroke-like episodes (MELAS) disease [105] and diabetes [106]. According to the established principle of RdRR in yeast mtDNA segregation [33], treating human heteroplasmic m.3243A > G primary fibroblast cells with hydrogen peroxide at an optimal ROS level, the promoted shift of mt-allele segregation towards wild-type and mutant mtDNA homoplasmy was observed [18]. The mechanism underlying ROS-stimulated mt-allele segregation towards homoplasmy in human cells is rolling-circle mtDNA replication, in which the amount of intact circular mtDNA molecules used as rolling-circle type replication templates is reduced, but the amount of mtDNA concatemers is elevated and newly synthesized mtDNA is concatemeric in hydrogen peroxide-treated human cells [18]. A newly developed system for the detection of mtDNA species based on Southern blotting after PFGE-coupled 2D gel electrophoresis has shown that ROS-triggered mt-allele segregation correlates with the production of mtDNA concatemers [18]. 

The ROS-stimulated mt-allele segregation via the rolling-circle mechanism raises possibility of restoring mtDNA homoplasmy from heteroplasmic human cells, without passage through the germline, by segregating mutant mtDNA molecules away from wild-type mtDNA copies within a cell; it could also be used to decrease the fraction of mutant mtDNA in a heteroplasmic cell during prenatal development [18]. Furthermore, this method also holds promise for screening iPS cells, or iPS cell-derived products, which have lower levels of point or deletion mutations in mtDNA (Figure 3). In addition, the RdRR mechanism first discovered in yeast may be universal among eukaryotes from yeast to humans [18]. We infer that the bootlace strand-asynchronous replication model, based on RITOLS, can occur using circular mtDNA as templates since intramolecular recombination converts concatemers to circular mtDNA molecules (also see: Figure 1). Of course, the proceeding of SCD replication also can occur on concatemers. 

## 10. Conclusions and Perspectives

Here, we introduced the rolling-circle type mtDNA replication mode driven by mitochondrial homologous recombination and described its potential importance for maintaining a healthy lifespan, preventing mitochondrial diseases, and understanding the nature of the mtDNA genetic bottleneck during oogenesis.

Rolling-circle mtDNA replication is initiated by a homologous DNA pairing protein, and thus has the ability to repair mtDNA DSBs by homologous DNA recombination, raising the possibility of preventing aging processes attributed to mtDNA deletions. The rolling-circle replication mode promotes the shift of mitochondrial alleles towards homoplasmy, and thus decreases or increases the proportion of mutant mtDNA at a single-cell level; this could yield a paradigm shift in the treatment of incurable mitochondrial diseases. Without the rolling-circle mode of mtDNA replication, it is not possible to fully explain the mtDNA genetic bottleneck. Recent findings regarding RdRR should provide an enthusiastic discussion within the field about mtDNA metabolism and inheritance.

## Figures and Tables

**Figure 1 genes-11-00514-f001:**
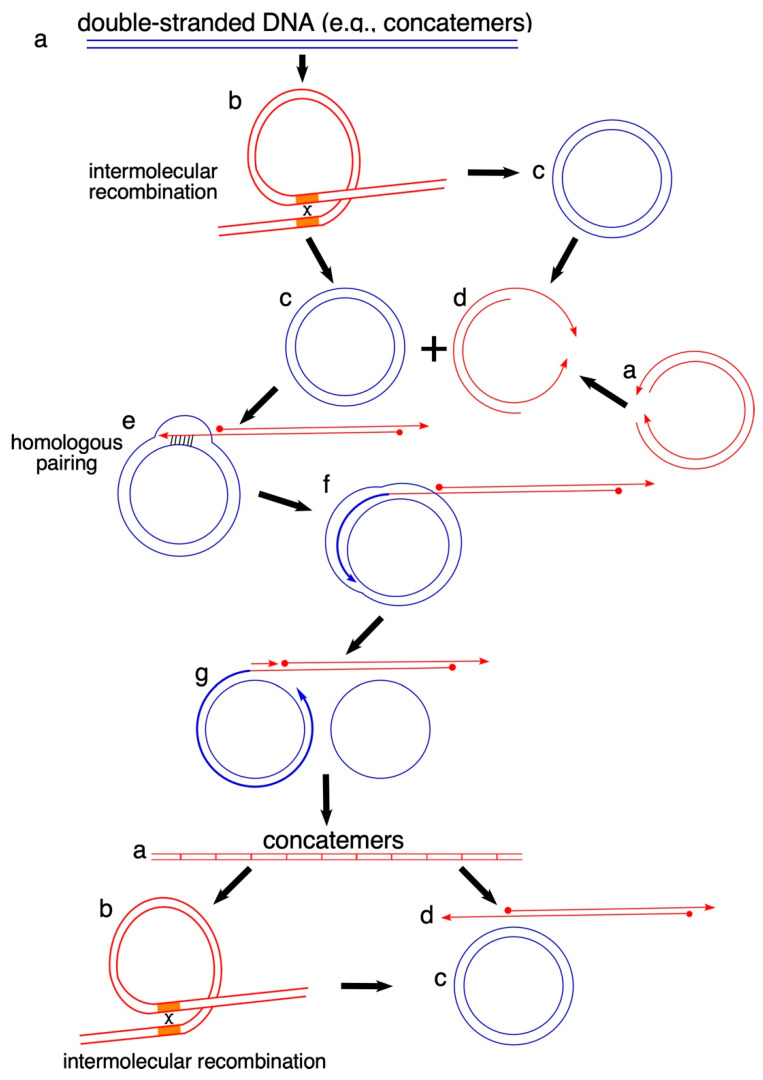
Homologous pairing–mediated mtDNA replication via a rolling-circle mechanism. mtDNA replication is initiated from double-stranded DNA, such as a concatemer (**a**). Intramolecular recombination (**b**) converts concatemers to circular mtDNA molecules (**c**). 5′→3′ exonuclease produces a 3′ single-stranded tail of linear double-stranded mtDNA, followed end resection at DSBs (**c**–**d**). Homologous DNA recombinases such as Mhr1 initiate rolling-circle mtDNA replication in a heteroduplex joint (**e**), yielding replication intermediates (**e**–**g**) and products termed as concatemers, which are linear tandem multimers linked by head-to-tail unit-sized mtDNA (**a**). Intramolecular recombination (**b**) converts concatemers to circular mtDNA molecules (**a**), which may act as a template for rolling-circle mtDNA replication. Note: only circular mtDNA molecules (**c**) are resistant to degradation by exonuclease activities.

**Figure 2 genes-11-00514-f002:**
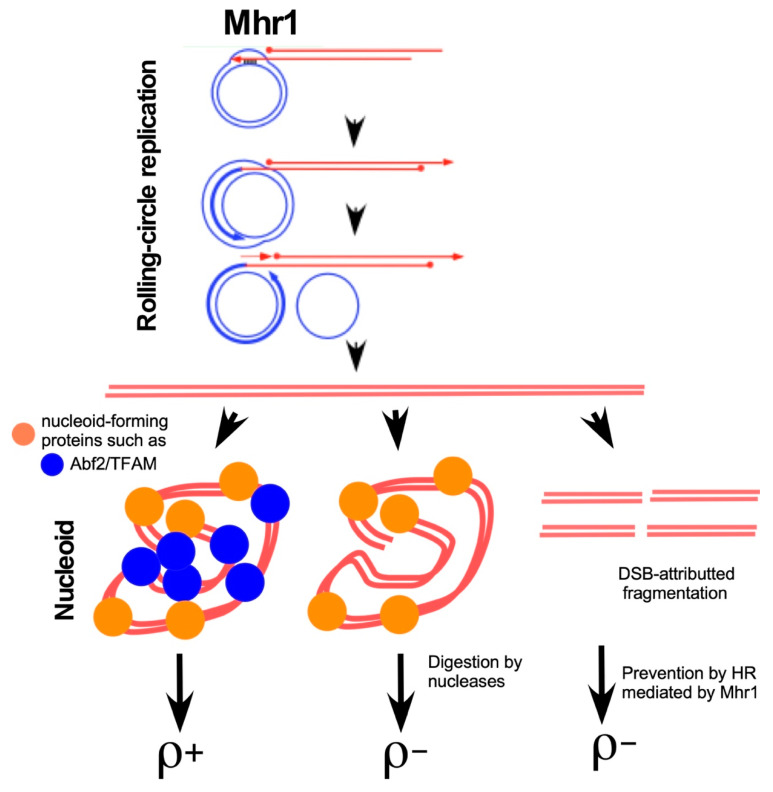
Packaging of mtDNA in nucleoids and DSB repair by Mhr1-mediated homologous recombination. The products of rolling-circle mtDNA replication, as linear mtDNA molecules, are packaged in mitochondrial nucleoids by nucleoid-forming proteins such as Abf2/TFAM for inheritance. Otherwise, they are susceptible to DNA damage and digestion by nucleases. mtDNA deletions caused by DSBs are primarily prevented by homologous recombination mediated by mitochondrial recombinases such as Mhr1. *ρ*+, respiration-proficient cells of *S. cerevisiae*. *ρ*−, respiration-deficient cells of *S. cerevisiae,* which contain mtDNA with a large deletion or tandem arrays of a mtDNA segment.

**Figure 3 genes-11-00514-f003:**
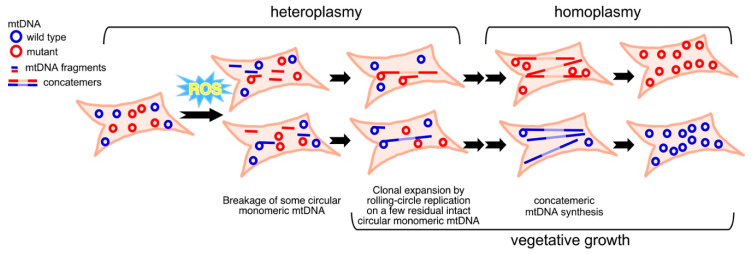
Restoration of homoplasmy from heteroplasmy through mitochondrial allele segregation stimulated by ROS. In hydrogen peroxide-treated MELAS cells, ROS cause partial breakage of intact circular monomeric mtDNA to decrease the number of templates. Concatemers synthesized by rolling-circle mtDNA replication, using residual whole monomeric mtDNAs as templates, allow restoration of homoplasmy during vegetative growth.

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
