# Peer review of "Rolling-Circle Replication in Mitochondrial DNA Inheritance: Scientific Evidence and Significance from Yeast to Human Cells"

_genes, 2020, doi:10.3390/genes11050514_

Round 1

Reviewer 1 Report

This is a concise review on the potential implication of the rolling-circle replication mode on mitochondrial DNA dynamics and possibly, mitochondrial function. It is thought-provocating and may be interesting for those in the field especially when dealing with cells under stress and specific pathophysiological conditions. I have only a few suggestions as listed below.

In the introduction, it may be worthwhile to briefly describe the current models of mtDNA replication in mammalian cells, and comment on the recently described Talon technology for removal of specific mtDNA. When discussing mtDNA recombination in yeast, the Rad52 homolog Mgm101 is missed out. This protein has been shown to play a role in homologous recombination.

Other comments:

Line 29, “ ………(mtDNA). mtDNA encodes …..” instead of “…….(mtDNA), which…..”.

Line 38, “incurable”, instead of “intractable”.

Line 46: “that mitochondria cells”?

Line 48: “the  -type mode”?

Line 71: “The main problem” for what? Something is missing here. Probably for inconsistency with rapid segregation?

Line 135: “parental gene”?

Line 200: “73”?

Line 205: “established” instead of “active”

Author Response

Reviewer #1 (Comments and Suggestions for Authors)

This is a concise review on the potential implication of the rolling-circle replication mode on mitochondrial DNA dynamics and possibly, mitochondrial function. It is thought-provocating and may be interesting for those in the field especially when dealing with cells under stress and specific pathophysiological conditions. I have only a few suggestions as listed below.

In the introduction, it may be worthwhile to briefly describe the current models of mtDNA replication in mammalian cells, and comment on the recently described Talon technology for the removal of specific mtDNA. When discussing mtDNA recombination in yeast, the Rad52 homolog Mgm101 is missed out. This protein has been shown to play a role in homologous recombination.

Other comments:

Line 29, “ ………(mtDNA). mtDNA encodes …..” instead of “…….(mtDNA), which…..”.

Line 38, “incurable”, instead of “intractable”.

Line 46: “that mitochondria cells”?

Line 48: “the  -type mode”?

Line 71: “The main problem” for what? Something is missing here. Probably for inconsistency with rapid segregation?

Line 135: “parental gene”?

Line 200: “73”?

Line 205: “established” instead of “active”

Response to Comments and Suggestions of Reviewer #1: 

: Thank you very much for your constructive suggestions.

Accordingly, we briefly described the current models of mtDNA replication in mammalian cells.

In mammalian cells, mtDNA replication mechanisms contain the bootlace strand-asynchronous replication model, based on the mechanism of ribonucleotide incorporation throughout the lagging strand (RITOLS) mtDNA replication and strand-coupled DNA replication (SCD replication) model have been proposed, and have been well described in a review article. We added the description on page 4, lines16 and page 5, lines1-5. The cited references are from Ian Holt’s group.

We also commented on the recently described TALEN technology for the removal of specific mtDNA.

In recent years, the expression of mitochondria-targeted transcription activator-like effector nucleases (TALENs), to cleave pathogenic mtDNA mutations raises the possibility that these mitochondrial nucleases can be therapeutic for some mitochondrial diseases, but it is essential to introduce exogenous factors into heteroplasmic cells for the removal of specific mtDNA. We added our comments on page 4, lines 1-6. The cited references are from Morae’s group.

We further described the function of the Rad52 homolog Mgm101, which plays a critical role in homologous recombination, but was missed out in the previous version.

A mitochondria-localized Rad52-related protein Mgm101 has a short carboxyl-terminal tail for single-stranded DNA binding is required for mitochondrial DNA recombination to maintain yeast mtDNA on page 12, lines 5-7. The cited references are from Chen’s group. 

We made corrections for other comments:

Line 29, “ ………(mtDNA). mtDNA encodes …..” instead of “…….(mtDNA), which…..”.

Accordingly, we made a change on page3, line 4.

Line 38, “incurable”, instead of “intractable”.

Accordingly, we used “incurable,” instead of “intractable” on page 2, line 14, page 3, line14, and page 15, line 15.

Line 46: “that mitochondria cells”?

We left “that mitochondria,” but deleted “cells” on page 4, line 11.

Line 48: “the  -type mode”?

We added q before “-type mode” on page 4, line 13.

Line 71: “The main problem” for what? Something is missing here. Probably for inconsistency with rapid segregation?

Here should be for the explanation of the rapid segregation of mt-alleles towards homoplasmy on page 6, lines 12-13.

We made a change.

Line 135: “parental gene”?

Here should be the wild-type gene on page 11, line 5.

We made a change.

Line 200: “73”?

“73” in the previous manuscript is the cited reference # 83 in the revised manuscript, which showed ∆abf2 lose mtDNA phenotype due to mtDNA deletions.

Line 205: “established” instead of “active”

Accordingly, we used “established” instead of “active” on page 16, line 5.

Reviewer #2 (Comments and Suggestions for Authors)

Dear Feng Ling and Minoru Yoshida,

I read your review entitled "Rolling-circle replication in mitochondrial DNA inheritance: Scientific evidence and significance from yeast to human cells" with interest as the mechanism of mitochondrial DNA replication is still under discussion.

I feel however that the review requires some editing.

To make the interesting topic more accessible to a wider readership, I would have the following suggestions:

Start the review with the different types of mitoDNA structures that are found in yeast, C.elegans and human cells as this throws up the question of the underlying replication mechanisms.

Then, introduce the problem of heteroplasmy and homoplasmy and how it relates to disease (include at least one disease example) and ROS load.

This could be followed by an overview of the suggested mitoDNA replication models: Rolling-circle replication, Bootlace strand-asynchronous replication (RITOLS), Strand-coupled DNA replication (SCD replication) and recombination-driven DNA replication in the context of the observed DNA structures and replication intermediates.

The review may now end with your arguments in favor of a rolling-circle model especially under high ROS load and in disease situations. The role of recombination (e.g. Mhr1) should be discussed in the context of pathway choice between the rolling circle and the strand-coupled DNA replication which both could produce concatemers and may explain the low amount of circular mitoDNA for example in S.cerevisiae petite mutants.

Response to Comments and Suggestions of Reviewer #2: 

Thank you very much for your constructive suggestions regarding the context of our manuscript.

We edited the manuscript accordingly. After reading the suggestions of the reviewer, we realized that the recombination-driven rolling-circle mtDNA replication(RdRR) occurs ahead of the mtDNA replication mechanisms observed in mammalian cells, especially under higher ROS load as newly synthesized mtDNA is concatemeric in hydrogen peroxide-treated human cells.
 We added the description on page 2, lines 9-12.

We infer that the bootlace strand-asynchronous replication, based on the RITOLS model, can occur using circular mtDNA as templates since intramolecular recombination converts concatemers to circular mtDNA molecules and the proceeding of SCD replication can occur on concatemers. We added the description on page 17, lines 9-12.

Accordingly, we briefly described the current models of mtDNA replication in mammalian cells, which was also recommended by reviewer #1. Namely, in mammalian cells, mtDNA replication mechanisms contain the bootlace strand-asynchronous replication model, based on the mechanism of ribonucleotide incorporation throughout the lagging strand (RITOLS) mtDNA replication and strand-coupled DNA replication (SCD replication) model have been proposed, and have been well described in the review article written by Dr. T. Yasukawa. We added the description on page 4, line 16 and page 5, lines1-5.

In mammalian cells, exonucleases rapidly degrade linear mtDNA, such as concatemers synthesized via strand-coupled DNA replication (SCD replication); only the remaining circular mtDNA molecules can be observed by microscopy. The rolling circle replication can sustainably produce linear tandem multimers linked by head-to-tail unit-sized mtDNA, termed mtDNA concatemers using circular mtDNA molecules resistant to the degradation by exonuclease as templates, but the SCD replication can’t.  Accordingly, we added the description to discuss in the context of pathway choice between the rolling circle and the strand-coupled DNA replication on page 8, lines 7-13.

In S.cerevisiae petite mutants, mtDNA deletion-attributed dysfunctional mitochondria can serve as a signaling platform to promote the loss of redox homeostasis and ROS accumulation, therefore more concatemers accumulate likely by ROS-activated rolling-circle replication. We added the description on page 8, line 16 and page 9, lines 1-3.

Reviewer 2 Report

Dear Feng Ling and Minoru Yoshida,

I read your review entitled "Rolling-circle replication in mitochondrial DNA
inheritance: Scientific evidence and significance from yeast to human cells" with interest as the mechanism of mitochondrial DNA replication is still under discussion.

I feel however that the review requires some editing.

To make the interesting topic more accessible to a wider readership, I would have the following suggestions:

Start the review with the different types of mitoDNA structures that are found in yeast, C.elegans and human cells as this throws up the question of the underlying replication mechanisms.

Then, introduce the problem of heteroplasmy and homoplasmy and how it relates to disease (include at least one disease example) and ROS load.

This could be followed by an overview of the suggested mitoDNA replication models: Rolling-circle replication, Bootlace strand-asynchronous replication (RITOLS), Strand-coupled DNA replication (SCD replication) and recombination-driven DNA replication in the context of the observed DNA structures and replication intermediates.

The review may now end with your arguments in favour of a rolling-circle model especially under high ROS load and in disease situations. The role of recombination (e.g. Mhr1) should be discussed in the context of pathway choice between the rolling circle and the strand-coupled DNA replication which both could produce concatemers and may explain the low amount of circular mitoDNA for example in S.cerevisiae petite mutants.

Many of these elements are in the current manuscript but appear at different places making it rather difficult for the reader to follow the main argument.

Author Response

Reviewer #2 (Comments and Suggestions for Authors)

Dear Feng Ling and Minoru Yoshida,

I read your review entitled "Rolling-circle replication in mitochondrial DNA inheritance: Scientific evidence and significance from yeast to human cells" with interest as the mechanism of mitochondrial DNA replication is still under discussion.

I feel however that the review requires some editing.

To make the interesting topic more accessible to a wider readership, I would have the following suggestions:

Start the review with the different types of mitoDNA structures that are found in yeast, C.elegans and human cells as this throws up the question of the underlying replication mechanisms.

Then, introduce the problem of heteroplasmy and homoplasmy and how it relates to disease (include at least one disease example) and ROS load.

This could be followed by an overview of the suggested mitoDNA replication models: Rolling-circle replication, Bootlace strand-asynchronous replication (RITOLS), Strand-coupled DNA replication (SCD replication) and recombination-driven DNA replication in the context of the observed DNA structures and replication intermediates.

The review may now end with your arguments in favour of a rolling-circle model especially under high ROS load and in disease situations. The role of recombination (e.g. Mhr1) should be discussed in the context of pathway choice between the rolling circle and the strand-coupled DNA replication which both could produce concatemers and may explain the low amount of circular mitoDNA for example in S.cerevisiae petite mutants.

Response to Comments and Suggestions of Reviewer #2: 

Thank you very much for your constructive suggestions regarding the context of our manuscript.

We edited the manuscript accordingly. After reading the suggestions of the reviewer, we realized that the recombination-driven rolling-circle mtDNA replication(RdRR) occurs ahead of the mtDNA replication mechanisms observed in mammalian cells, especially under higher ROS load as newly synthesized mtDNA is concatemeric in hydrogen peroxide-treated human cells.
 We added the description on page 2, lines 9-12.

We infer that the bootlace strand-asynchronous replication, based on the RITOLS model, can occur using circular mtDNA as templates since intramolecular recombination converts concatemers to circular mtDNA molecules and the proceeding of SCD replication can occur on concatemers. We added the description on page 17, lines 9-12.

Accordingly, we briefly described the current models of mtDNA replication in mammalian cells, which was also recommended by reviewer #1. Namely, in mammalian cells, mtDNA replication mechanisms contain the bootlace strand-asynchronous replication model, based on the mechanism of ribonucleotide incorporation throughout the lagging strand (RITOLS) mtDNA replication and strand-coupled DNA replication (SCD replication) model have been proposed, and have been well described in the review article written by Dr. T. Yasukawa. We added the description on page 4, line 16 and page 5, lines1-5.

In mammalian cells, exonucleases rapidly degrade linear mtDNA, such as concatemers synthesized via strand-coupled DNA replication (SCD replication); only the remaining circular mtDNA molecules can be observed by microscopy. The rolling circle replication can sustainably produce linear tandem multimers linked by head-to-tail unit-sized mtDNA, termed mtDNA concatemers using circular mtDNA molecules resistant to the degradation by exonuclease as templates, but the SCD replication can’t.  Accordingly, we added the description to discuss in the context of pathway choice between the rolling circle and the strand-coupled DNA replication on page 8, lines 7-13.

In S.cerevisiae petite mutants, mtDNA deletion-attributed dysfunctional mitochondria can serve as a signaling platform to promote the loss of redox homeostasis and ROS accumulation, therefore more concatemers accumulate likely by ROS-activated rolling-circle replication. We added the description on page 8, line 16 and page 9, lines 1-3.

Round 2

Reviewer 2 Report

Dear authors,

thank you very much for responding so positively to my comments.

Here are my final comments:

Line 35: Homoplasmy of mtDNA, in which all mtDNA copies in one cell have identical sequences...

I still think that section 2 is still too short as it focuses on the Phi (Φ) replication, but the other models (Strand-Displacement Replication, RNA Incorporation, coupled leading & lagging strand synthesis, recombination-mediated replication) should also be briefly mentioned to give the reader the complete picture of the scientific question discussed. 

In section 4, the switch to the rolling circle replication model (line 105) is too fast and the significance of the petite phenotype in yeast is not yet well introduced. Petite needs to be explained as it is a very specific yeast mutation. Also the role of ROS in the rolling circle replication should be explained in more detail here.

Line 189 ATM-Chk2 kinase pathway better than ATM/Chk2 pathway as “/” indicates two alternative names for the same thing.

Legend Fig 2 p+ p- not explained

Author Response

Response to Additional Comments and Suggestions of Reviewer #2 

Thank you very much for your constructive suggestions.

 Line 35: Homoplasmy of mtDNA, in which all mtDNA copies in one cell have
identical sequences... I still think that section 2 is still too short as it focuses on the Phi (Φ) replication, but the other models (Strand-Displacement Replication, RNA Incorporation, coupled leading & lagging strand synthesis, recombination-mediated replication) should also be briefly mentioned to give the reader the complete picture of the scientific question discussed.

Accordingly, we described in more detail regarding the mtDNA replication models in mammals and added the description on page 1, lines 37-41, and page 2, lines 42-50.

The return of a heteroplasmic mtDNA mutation to homoplasmy occurs in only two or three generations of Holstein cows, but the underlying mechanism remains highly controversial. It is partially because the mechanism for mtDNA replication in mammals remains unclear. In mammals, a strand-displacement mechanism was proposed based on circular replicative intermediates observed under an electron microscope, which was further refined as a model for replication of animal mtDNA. So far, two asynchronous [strand displacement and ribonucleotide incorporation throughout the lagging strand (RITOLS)] and one synchronous (strand coupled) replication models have been proposed for human mtDNA replication. In the asynchronous models, replication from the origin in the H-chain starts earlier, so that the replication of the two chains ends at different times. The synchronous model is more traditional and implies two replication forks with leading and lagging strands initiated at the same origin. For details, please refer to the review article. Recently, the rolling-circle mtDNA replication mechanism in hydrogen peroxide-treated human cells, which promotes mt-allele segregation towards mt-homoplasmy, was revealed.

 In section 4, the switch to the rolling circle replication model (line 105) is too fast.

Accordingly, we introduced the DNA replication systems using the rolling-circle (s) mode. How concatemers can avoid degradation by exonucleases is an open issue.     

The major species of S.cerevisiae mtDNA are linear head-to-tail multimers of genomic unit DNA with variable sizes, termed concatemers. Concatemers can be mainly formed through rolling-circle DNA replication or homologous DNA recombination. Rolling-circle DNA replication is initiated through recombination-dependent mechanisms in some DNA replication systems, such as bacteriophage lambda at the late stage of infection, plasmids and even the chromosome in SOS-induced E. coli cells. In the later stages of l infection, the DNA replication switches from a theta (q) mode to a rolling-circle (s) mode, and this switch requires the proteins encoded by the redα (λ exonuclease) and redβ  protein) genes required for homologous DNA recombination. In phage T4 of E. coli, concatemers are formed through homologous DNA recombination. We added the description on page 3, lines 116-125.

The petite phenotype in yeast is not yet well introduced. Petite needs to be explained as it is a very specific yeast mutation.

Accordingly, we described the petite phenotype in yeast in more detail.

S.cerevisiae petite mutants are respiration-deficient cells, which are unable to grow on media containing only non-fermentable carbon sources (such as glycerol or ethanol) and form small colonies when grown in the presence of fermentable carbon sources (such as glucose), and contain mtDNA with a large deletion or tandem arrays of a mtDNA segment. We added the description on page 3, lines 127-131.

Also the role of ROS in the rolling circle replication should be explained in more detail here.
Accordingly, we explained the role of ROS in the rolling circle replication in more detail.

mtDNA deletion-attributed dysfunctional mitochondria can serve as a signaling platform to promote the loss of redox homeostasis and ROS accumulation. More accumulation of concatemers in yeast petite mutant cells is very likely that the excision-repair enzyme Ntg1 recognizes the bases oxidized by ROS and introduces a DSB at the mtDNA replication origin ori5 to initiate the rolling-circle mtDNA replication. We added the descriptions on page 3, lines 131-135.

We also explained the role of ROS in the rolling circle replication in more detail below.

In S. cerevisiae, the oxidized base excision-repair enzyme Ntg1 introduces a DSB in the single-stranded regions at the mtDNA replication origin ori5; this DSB initiates the rolling-circle mtDNA replication mediated by the homologous DNA pairing protein Mhr1. Thus, optimal amounts of ROS promote mt-allele segregation mediated concatemers produced by the rolling-circle mtDNA replication. We added the description on page 5, lines 192-198.

Line 189 ATM-Chk2 kinase pathway better than ATM/Chk2 pathway as “/” indicates two alternative names for the same thing.

Accordingly, we used the “ATM-Chk2 kinase pathway” instead of the “ATM/Chk2 pathway” on page 6, line 216.

Legend Fig 2 ρ+ ρ- not explained.

Accordingly, we added the explanation on ρ+ and ρ- on page 6, lines230-232.

ρ+, respiration-proficient cells of S. cerevisiae. ρ-, respiration-deficient cells of S. cerevisiae, which contain mtDNA with a large deletion or tandem arrays of a mtDNA segment.
